# Targeted Sequencing of Pancreatic Adenocarcinomas from Patients with Metachronous Pulmonary Metastases

**DOI:** 10.3390/genes11121391

**Published:** 2020-11-24

**Authors:** Viktor Hlavac, Beatrice Mohelnikova-Duchonova, Martin Lovecek, Jiri Ehrmann, Veronika Brynychova, Katerina Kolarova, Pavel Soucek

**Affiliations:** 1Biomedical Center, Faculty of Medicine in Pilsen, Charles University, 306 05 Pilsen, Czech Republic; veronika.brynychova@szu.cz (V.B.); pavel.soucek@szu.cz (P.S.); 2Toxicogenomics Unit, National Institute of Public Health, 100 42 Prague, Czech Republic; 3Department of Oncology, Institute of Molecular and Translational Medicine, Faculty of Medicine and Dentistry, Palacky University, 779 00 Olomouc, Czech Republic; d.beatrice@seznam.cz (B.M.-D.); kolaka03@gmail.com (K.K.); 4Department of Surgery I, University Hospital Olomouc and Faculty of Medicine and Dentistry, Palacky University, 779 00 Olomouc, Czech Republic; mlovecek@seznam.cz; 5Department of Clinical and Molecular Pathology, University Hospital Olomouc and Faculty of Medicine and Dentistry, Palacky University, 779 00 Olomouc, Czech Republic; jiri.ehrmann@gmail.com

**Keywords:** pancreas, adenocarcinoma, next-generation sequencing, survival, pulmonary, metastases

## Abstract

Mutation spectra of 250 cancer driver, druggable, and actionable genes were analyzed in surgically resected pancreatic ductal adenocarcinoma (PDAC) patients who developed metachronous pulmonary metastases. Targeted sequencing was performed in DNA from blood and archival samples of 15 primary tumors and three paired metastases. Results were complemented with the determination of G12V mutation in *KRAS* by droplet digital PCR. The median number of protein-changing mutations was 52 per patient. *KRAS* and *TP53* were significantly enriched in fractions of mutations in hotspots. Individual gene mutation frequencies or mutational loads accounting separately for drivers, druggable, or clinically actionable genes, did not significantly associate with patients’ survival. *LRP1B* was markedly mutated in primaries of patients who generalized (71%) compared to those developing solitary pulmonary metastases (0%). *FLG2* was mutated exclusively in primary tumors compared to paired metastases. In conclusion, signatures of prognostically differing subgroups of PDAC patients were generated for further utilization in precision medicine.

## 1. Introduction

Pancreatic ductal adenocarcinoma (PDAC, Online Mendelian Inheritance in Man–OMIM no. 260350) represents the 14th most frequent malignant diagnosis with more than 432,000 predicted deaths in 2018 worldwide [1]. More importantly, PDAC mortality is projected to become the second leading cause of cancer-related death in the USA by 2030 [2]. Due to the delayed manifestation of symptoms, PDAC is usually diagnosed too late and only about 10–20% of patients can be surgically treated [3]. Additionally, PDAC has generally a very poor prognosis due to the lack of long-term response to the systemic chemotherapy, with a median survival of 6.8–16.4 months in metastatic patients in palliative setting [4] and 25.5–54 months in resected and adjuvantly treated patients [5,6]. Moreover, recent data suggest that the prognosis of patients relapsing after surgery of the primary tumor site further differs according to the organ where metastasis develops first and whether it is accompanied by generalization to other organs.

The liver and lungs represent the most frequent sites of secondary metastasis of PDAC. Interestingly, long-term surviving patients, a very rare fraction of PDAC, have more frequently metastases only in the lungs [7,8]. Optimal systemic chemotherapy may further prolong their life expectancy. Patients with isolated metachronous pulmonary metastases may even benefit from surgical treatment [9]. Although several studies reported the genomic landscape of unselected PDAC primary tumors [10,11,12,13] and even its genealogy through profiling of primary tumor and distant metastasis samples [14,15], we still lack reliable information about underlying mechanisms of pulmonary metastasis in PDAC. Such information may be derived either from primary tumors of patients who later relapse in the lungs or by comparing their primaries with secondary pulmonary tissues. However, access to the secondary metastases is, needless to say, very limited. Nevertheless, knowledge of the molecular and particularly genetic background of this process would allow identification of actionable genomic alterations [16], development of targeted therapeutics, and discovery of biomarkers of site-specific metastasis, which could be traceable, e.g., by liquid biopsy.

The present study aimed to analyze somatic mutational spectra of a selected gene panel in the retrospectively collected cohort of surgically resected PDAC patients with curative intent who all subsequently relapsed with either solitary metachronous pulmonary metastases or metastases in lungs together with other organs. Genetic alterations depicting the PDAC pulmonary metastasis process associated with the long-term survival of patients may provide a lead for follow-up studies targeted to utilizing this somatic profile in precision medicine.

## 2. Materials and Methods

### 2.1. Patients

Samples of surgically resected, chemoradiation naïve, primary PDAC tumors were retrospectively collected from 15 PDAC patients who developed pulmonary metastases in the course of their disease (P1–15). Patients were diagnosed and treated in University Hospital in Olomouc between 2008 and 2016. Paired samples of pulmonary metastases were available for three of those 15 patients (M1–3) and blood DNA was collected from 12 patients with pulmonary PDAC metastases for the creation of the panel of normal variants used for method adjustment during the bioinformatics evaluation. Four blood samples were paired to the abovementioned patients (matched-normal blood or tissue was not available for the rest of the archival samples).

The clinical data, including age, gender, date of diagnosis, stage, the histological type and grade of the tumor, lymphatic, vascular and perineural invasion, data concerning surgery and oncological treatment, treatment outcomes, date and site of recurrence, and overall survival (OS), were obtained from medical records. The OS was defined as the time elapsed between surgical resection and death of any cause or patient censoring. The main clinical characteristics of the whole group are summarized in Table 1. None of the patients had received neoadjuvant chemo(radio)therapy.

All procedures performed in this study followed the ethical standards of the Institutional Review Board of the University Hospital in Olomouc, Czech Republic, (approval reference no. 159/16) and with the 1964 Helsinki declaration and its later amendments or comparable ethical standards. The experimental protocol of this study was also approved by the Institutional Review Board of the University Hospital in Olomouc, Czech Republic, (approval reference no. 159/16). This article does not contain any studies with animals performed by any of the authors.

### 2.2. DNA Isolation and Quantification

DNA from the archival formalin-fixed paraffin-embedded (FFPE) blocks was isolated with the RecoverAll Total Nucleic Acid Isolation Kit (Ambion/Thermo Fisher Scientific, Prague, Czech Republic). The first layer of tumor blocks was discarded, and the remaining sample was macrodissected to remove as much normal tissue as possible before purifying DNA. In total, eight 10-µm-thick sections were cut from each tumor block, placed into sterile 1.5-mL centrifuge tubes, and processed according to the manufacturer’s protocol. All DNA samples were treated by RNase during purification. DNA was eluted to 60 µL of nuclease-free water and stored at −20 °C until further use. DNA from peripheral blood lymphocytes was isolated and stored according to the published procedure [17].

DNA was quantified using Quant-iT PicoGreen dsDNA Reagent and Kits (Invitrogen/Thermo Fisher Scientific, Waltham, MA, USA) and plate reader Infinite 200 (Tecan Group Ltd., Männedorf, Switzerland). DNA purity, such as A260/280 and A260/230 ratios, was checked by the NanoDrop Spectrophotometer 2000 (Thermo Scientific/Thermo Fisher Scientific, Waltham, MA, USA). DNA quality was further checked by the amplification of two fragments (117 bp and 382 bp in length) of the β-actin gene (ACTB; OMIM no. 102630) using the polymerase chain reaction (PCR). PCR reaction was carried out in a 10-μL reaction mixture containing 1 × HOT FIREPol Blend Master Mix 12.5 mM Ready to Load (Solis BioDyne, Tartu, Estonia), 5 pmol each forward and reverse primer of 117 bp fragment (F: 5′-CTGGCACCACACCTTCTACA-3′, R: 5′-CCACTCACCTGGGTCATCTT-3′) or 382 bp fragment (F: 5′-CTGGCACCACACCTTCTACA-3′, R: 5′-GCTTTACACCAGCCTCATGG-3′) (Thermo Fisher Scientific, Waltham, MA, USA), and nuclease-free water. Cycling parameters were: initial denaturation at 94 °C for 3 min, followed by 35 cycles consisting of denaturation at 94 °C for 30 sec, annealing at 60 °C for 30 sec, elongation at 72 °C for 60 sec, and followed by final elongation at 72 °C for 10 min. PCR products were checked on 2% agarose gel electrophoresis. The 117-bp fragment was detected in all DNA samples (100%) and the 382-bp fragment in 11 samples (61%). Therefore, a method based on the amplification of short fragments was used for subsequent library preparation.

### 2.3. Design of Gene Panel for Targeted Sequencing

For the study, the gene panel was designed to contain 20 most frequently mutated genes in PDAC and 20 top mutated genes in lung adenocarcinoma (LUAD) besides those selected for PDAC. Genes were selected using the Catalogue of Somatic Mutations in Cancer database (COSMIC, https://cancer.sanger.ac.uk/cosmic) in 2018. Further, 210 driver and druggable genes (defined as druggable genome and clinically actionable in the Drug Gene Interaction Database (DGIdb, http://dgidb.org/) and genes representing, e.g., RAS, Wnt, cadherin, epidermal growth factor receptor (EGFR, OMIM: 131550), p53, angiogenesis, DNA repair, and inflammatory signaling pathways, were added to the panel (*n* = 250 genes, Appendix A) based on literature search [10,11,12,13,14,15]. Target enrichment probes were designed in the NimbleDesign web tool (Nimblegen-Roche, Madison, WI, USA) encompassing all coding exons and untranslated regions (UTR) of genes of interest.

### 2.4. Library Preparation and DNA Sequencing

The libraries were prepared from tumor and blood DNA using SeqCap EZ choice (Nimblegen-Roche, Prague, Czech Republic), according to the manufacturer’s protocol as described previously [18]. Briefly, 500 ng (tumor) or 100 ng (blood) of DNA were fragmented using the Covaris M220 instrument (Covaris, Inc. Woburn, MA, USA) and used in the reaction. Tumor and blood libraries were sequenced on the HiSeq and MiSeq platforms, respectively (Illumina Inc., San Diego, CA, USA).

### 2.5. KRAS G12V Mutation Detection by Digital Droplet PCR

Digital droplet PCR (ddPCR) was performed using 2× ddPCR Supermix for Probes (without deoxyuridine triphosphate, Bio-Rad, Hercules, CA, USA), 20× ddPCR Mutation Detection Assay *KRAS* p.G12V c.35G > T (validated, ID: dHsaMDV2510592) (Bio-Rad, Hercules, CA, USA), the restriction enzyme HaeIII (4 U per reaction, prediluted in 10× CutSmart Buffer according to the manufacturer’s instructions) (New England Biolabs, Ipswich, MA, USA), and 20 ng DNA in a total reaction volume of 20 µL. DNA from PaCa-44, a human pancreatic ductal adenocarcinoma *KRAS* G12V heterozygous cell line, was used as a positive control. Three DNA samples (*KRAS* wild type) were obtained from peripheral blood lymphocytes of healthy donors and used as negative controls. Non-template control containing water added to the reaction mixture instead of DNA was also analyzed to control potential carryover contamination. All samples, including controls, were assayed in duplicates. The ddPCR was performed using the QX200 Droplet Digital PCR System (Bio-Rad, Hercules, CA, USA), according to the manufacturer’s recommendations. Cycling conditions were: denaturing at 95  °C for 10 min, 40 cycles of PCR at 94 °C for 30 s and 60 °C for 1 min, and a final extension at 98  °C for 10 min. The droplets from each well were analyzed according to the Poisson distribution and absolute quantifications of mutant and wild-type alleles were estimated using QuantaSoft Analysis Software version 1.7.4 (Bio-Rad, Hercules, CA, USA). No false-positive (mutant) droplets were detected in negative controls. The limit of detection for mutated allele was set to 0.1%. Copy numbers per reaction were used for the calculation of mutant allele frequency (mutated/(mutated + wild type) × 100).

### 2.6. Bioinformatics Analysis

Adapter sequences and low-quality bases were removed from raw FASTQ data using Trimmomatic 0.39 [19]. Reads were aligned to hg38 human reference genome using Burrows-Wheeler Aligner 0.7.17 (BWA, Cambridge, UK) [20]. Sequence alignment/map (SAM) to binary alignment map (BAM) format conversion was performed using Samtools 1.9 (Wellcome Trust Sanger Institute, Hinxton, UK). Duplicate removal was done using Picard 2.21.8. Base recalibration was done using Genome Analysis Toolkit (GATK) 4.1.3.0 (Broad Institute, Cambridge, UK) [21] and somatic variant identification was performed using Mutect2 tool according to GATK Best Practices. To account for Czech population-specific genetic variants, the national database of the National Center for Medical Genomics (https://ncmg.cz/en) was utilized during base recalibration. A panel of normals was created from 12 blood samples of the PDAC patients and samples were analyzed in pairs where applicable (*n* = 4). Found variants were annotated and converted to mutation annotation format (MAF) using Variant effect predictor 98.3 (VEP, Wellcome Trust Sanger Institute) [22] and vcf2maf, or Funcotator (GATK). Prediction of driver genes was performed in OncodriveCLUST [23] and Mutation Significance Covariates (MutSigCV) v1.3 [24]. Results from The Cancer Genome Atlas (TCGA, https://www.cancer.gov/tcga) data analyzed by MutSig2CV v3.1 in a cohort of PDAC (http://dx.doi.org/10.7908/C1513XNS) [25] and LUAD (http://dx.doi.org:/10.7908/C17P8XT3) [26] patients were obtained from Broad Firehose using firehose_get utility (analysis stamp: 2016_01_28, ref. (Broad_Institute_TCGA_Genome_Data_Analysis_Center, 2016a, b)). All open-source bioinformatics tools were obtained from Bioconda [27]. Plots were generated in R package Maftools [28]. A review of aligned reads was performed by Integrative Genomics Viewer (IGV) [29]. For the prediction of cancer driver genes, *p*-values were adjusted with Benjamini–Hochberg false discovery rate method [30] and *q* ≤ 0.1 was considered significant.

BAM files mapped to reference sequence were submitted to The National Center for Biotechnology Information (NCBI) Sequence Read Archive (SRA) database (https://www.ncbi.nlm.nih.gov/sra) under the accession number of BioProject: PRJNA631881.

### 2.7. Survival Analysis

Patients with OS exceeding two years since surgical removal of the tumor were considered as long-term survivors (*n* = 9) and those surviving shorter periods represented short-term survivors (*n* = 6). Differences in mutation load in selected genes between patients, divided by the survivor status (short vs. long), were compared by the Chi-square test. Survival functions for all patients divided by the mutation load were plotted by the Kaplan–Meier method and significance was calculated by the Log Rank test. A *p*-value of less than 0.05 was considered significant. Survival analyses were evaluated by the SPSS v16 program (SPSS Inc., Chicago, IL, USA).

## 3. Results

### 3.1. General Descriptors of Panel Sequencing

The average coverage of sequenced regions in tumor samples (*n* = 18) was 346.1 ± 219.8 (median 280.5). Of bases, 98.8% were covered at least 30×, and 97.1% of bases were covered at least 100×. The average coverage of sequenced regions in blood samples was 62.0 ± 13.0 (median 68.0) and 97.8% bases were covered at least 10×.

The total number of true somatic variants per sample was 254.5 ± 169.3 on average (ranging from 86 to 597, median 214). True somatic variants were then annotated and transformed to MAF. Funcotator annotated variants are presented in Appendix A. The total number of variants used in the following analyses was 1270 (*n* = 18 samples, mean 70.6 per sample, median 48.5 per sample) and 3412 silent and noncoding variants were excluded.

### 3.2. Discovery and Prediction of Variants in Primary Tumor Samples

The distribution of somatic variants in the 15 samples of primary tumors is depicted in Figure 1. In these primary tumors, the median number of variants per sample was 52. The most common class of variants was a missense mutation (Figure 1A,D,E) and the most frequent type was single nucleotide polymorphism (SNP, Figure 1B). The most common nucleotide change was C to T transition (Figure 1C). The top mutated gene was titin (*TTN*, OMIM no. 188840; Figure 1F) with 80 mutation events (Figure 1).

Next, the variants were filtered for the genes listed in the FrequentLy mutAted GeneS (FLAGS) database [31]. The top 20 genes scored by FLAGS were excluded from further analyses (Appendix A). Oncoplot of the list of genes with highest counts of variants in primary tumors is shown in Figure 2A. (Samples were sorted according to their OS (Figure 2B) and patients with OS longer than two years are on the right.)

We also performed the OncodriveCLUST algorithm for the prediction of driver genes. *KRAS* and *TP53* were significantly enriched in fractions of mutations in hotspots (*q* = 4.6 × 10^−6^ and *q* = 3.3 × 10^−3^, respectively). Mutations in *KRAS* and *TP53* genes in the primary tumors of 15 patients assessed by sequencing are summarized in Appendix A. Comparison of results of ddPCR for the most frequent *KRAS* G12V mutation with sequencing is in Table 2.

The ddPCR revealed *KRAS* G12V mutations in 10 patients (67%) and five patients were classified as non-carriers. Sequencing failed to detect this mutation in all samples with lower than 5% allele frequency (*n* = 5). In some of these patients, different mutations were observed (e.g., *KRAS* G12D). Thus, a combination of ddPCR and sequencing allowed the detection of *KRAS* mutations in 12 patients (80%). Despite that variants were not called by Mutect2, the inspection of aligned reads in IGV revealed alternative bases in tumor cells in three cases with allele frequency over 1% (Appendix A). These alleles were originally not called due to the low quality of bases (Phred score < 30). In the end, two samples with allele frequency lower than 1% were false negatively, assigned as wild type by sequencing. Thus, the reproducibility of the next-generation sequencing method is very good. Based on the ddPCR validation of G12V mutation, the minimal allele frequency cutoff of 1% was considered detectable. This value could be further extrapolated to all variants in general, although some manual curation may be necessary.

Mutations in *KRAS* or *TP53* did not significantly associate with the survival of patients (Figure 3A,B). The forest plot shows genes’ odds ratios (ORs) of long-term surviving patients compared to short-term survivors. Genes with *p* < 0.35, i.e., *NBEA*, *DNAH9*, *ZFHX4*, *KRAS*, *TP53*, and *MUC17,* are displayed. None of the genes significantly associated with survival (Figure 3C).

No genes were significantly enriched with mutations in 15 PDAC patients according to MutSigCV (*q* > 0.1, data not shown). On the other hand, 603 genes were significantly mutated in 126 PDAC patients and 54 genes were significantly mutated in 533 LUAD patients in studies of Broad Institute TCGA Genome Data Analysis Center (see section Bioinformatics Analysis). Our results compared with Broad Institute significantly mutated genes in PDAC patients in MutSig2CV (applicable for *n* = 23 genes) are presented in Appendix A and mutated genes in LUAD patients in MutSig2CV (applicable for *n* = 13 genes) are presented in Appendix A (−log10 *q*-values provided by MutSig2CV are shown on the right).

To compare the mutation load, we created a list of cancer driver genes and druggable or clinically actionable genes (Appendix A). Genes were selected as drivers if fulfilling one of the following criteria: (1) gene in a list of pan-cancer driver genes predicted by COSMIC Cancer Gene Census (https://cancer.sanger.ac.uk/census), (2) gene predicted as a driver in pan-cancer comparison or specifically in PDAC or LUAD according to Bailey et al. [32], or (3) genes predicted by MutSig2CV for PDAC or LUAD cohorts (see section Patients and Methods: Bioinformatics Analysis). Druggable or clinically actionable genes were selected using DGIdb (http://dgidb.org/). As a result, mutation load was evaluated in 72 cancer driver genes and separately in 129 druggable or clinically actionable genes, of which 59 overlapped with driver genes. Mutation load (either all variants or only functional variants, e.g., variants without noncoding and silent mutations) in these genes did not significantly differ in short-term surviving patients compared to long-term survivors (the Chi-square test *p* > 0.05). Survival functions of patients divided by the mutational load did not significantly differ as well (the Log Rank test *p* > 0.05). Similarly, survival functions or survivor status did not differ by the mutational load in FLAGS and the most mutated gene *TTN* (*p* > 0.05).

### 3.3. Variant Discovery and Comparison in Paired Tumor-Metastatic Tissues

We compared three sample pairs of primary tumors and matched pulmonary metastases. The summary of variant classes and counts is shown in Appendix A. Notably, the mutation spectra differed markedly between the primary tumors and metastases (Figure 4), e.g., *FLG2* was mutated in all primary tumors, but none of the corresponding metastases.

*ATM* and *GRIN2A* mutations were found in two metastases, but not detected in any of the paired primary tumors. However, in one patient the mutation load was higher in tumor sample (P2) than in metastasis (M2), while in two patients mutation load was slightly higher in metastases (M1 and M3) than in tumors (P1 and P3, Figure 4). Comparison of *KRAS* and *TP53* genes between primary tumors and metastases is shown in a lollipop plot in Figure 3D and for *KRAS* also in Table 2. All variants were present in both primary tumors and their paired metastases, except one primary tumor (P1), which did not harbor p.K132R in *TP53,* while this variant was present in the metastatic lesion (M1). Oncoplot with comparison to Broad Institute MutSig2CV mutated genes is shown in Appendix A.

### 3.4. Analysis of Differences among Patients with Different Metastatic Scenarios

When comparing the variant pattern in primary tumors of patients with solitary metachronous pulmonary oligometastases (*n* = 3, Figure 5A) with patients with the generalized disease (pulmonary metastases accompanied by other metastases in at least three different organs, *n* = 7, Figure 5B) no obvious difference was found.

In general, patients with solitary pulmonary metastases had longer OS (all survived longer than two years) and harbored a lower number of variants compared with the rest of the patients. Interestingly, *LRP1B*, a gene specifically deleted in non-small cell lung cancer (NSCLC), was mutated in 5/7 patients with the generalized disease (71%), but in none of the patients with solitary metastases (*n* = 3).

## 4. Discussion

This study analyzed the genetic profile of primary and metastatic lesions from PDAC patients with special attention to the patients’ survival and sites of first-detected relapse to reveal unique signatures of these prognostically differing subgroup(s) of patients for further disease management. Patients were collected retrospectively with no other criteria than subsequent development of metachronous metastatic lesions either isolated or accompanied with other metastases [9]. We observed a higher predominance of females in the set. This observation has already been reported [7].

Overall data analysis revealed that the average number of true somatic variants was 254, with a range between 86 and 597 per sample. Out of these, on average, 52 variants per sample were changing the protein sequence. Subsequent analysis using robust bioinformatics tools enabled thorough prioritization of the observed variants and genes. Less likely disease-associated genes were removed using FLAGS [31], and subsequent Oncoplot analysis identified *MUC5AC*, *FLG2, KMT2C, TP53, ASPM, KRAS, ZFHX4,* and *LRP1B* as the most frequently mutated genes (in more than 50% of primary tumors). *KRAS* (OMIM no. 190070) and *TP53* (OMIM no. 191170) were significantly enriched in fractions of mutation hotspots in driver genes by OncodriveCLUST. However, mutation frequencies of these genes did not significantly differ between groups of patients divided by survival and did not associate with survival functions as well.

Besides the above funnel-like prioritization approach considering the functional and causality predictions, we also applied a more general and less restricted search for determinants of long survival of PDAC patients. First, we reconfirmed the previously selected list of cancer drivers, druggable, and clinically actionable genes using publicly available databases and literature (COSMIC, DGIdb, ref. [32]). Then, mutational loads in these genes were calculated, in agreement with the previously optimized cutoff [33] and used for comparison of distributions of long- and short-term survivors and survival functions of patients. For this purpose, both overall and functionally relevant mutation loads predicted by the most frequently used tools (Funcotator and VEP) were used. No significant differences were observed by this approach either. Taken together, it was impossible to genetically discern long- and short-term survivors among PDAC patients by the sequencing of coding sequences of the selected genes.

Considering major PDAC driver genes, data supporting the association between somatic mutations and survival of PDAC patients exist. *KRAS* G12V mutation was associated with a shorter OS in a previous study on 73 PDAC patients [34]. Additionally, plasma G12V mutations in *KRAS* were associated with shorter OS and progression-free survival (PFS) of 113 Chinese PDAC patients and validated on a further 44 patients [35]. Interestingly, in a study on 177 resected PDAC patients, patients with somatic mutations in *KRAS* or *CDKN2A/p16* showed significantly shorter OS than patients with the wild-type genes [36]. Analogously, a combination of *KRAS* and *SMAD4* mutations in 74 PDAC patients was associated with shorter PFS in another study [37]. Biallelic loss of *TP53* was associated with worse OS and PFS in a whole-genome sequencing study on PDAC patients, although this association was not confirmed in multivariate analysis [38]. In a recent study, mutations in *TP53* were associated with shorter PFS in 101 PDAC patients [39]. In contrast with the present study, also data in two TCGA datasets available in cBioPortal [40,41] suggest the prognostic role of mutations in PDAC drivers *KRAS* and *TP53* (Appendix A).

In the present study, mutations in *LRP1B* (low-density lipoprotein receptor-related protein 1B, OMIM no. 608766) were detected in the majority of primary tumors from patients with the generalized disease (pulmonary and at least two other systems affected by metastasis), but not in any of the samples from patients with solitary pulmonary metastases. *LRP1B* is a very large gene [42], frequently mutated in human cancers. Its protein product is a putative tumor suppressor demonstrated to take part in the control of cell spreading, migration, and invasion, probably through regulating Rho family proteins [43]. Interestingly, *LRP1B* is among the recently suggested five genes representing a signature with prognostic potential in patients with stage III colon cancer [44]. Additionally, somatic mutations in *LRP1B* correlate with tumor mutation burden status in several cancers, e.g., nasopharyngeal carcinoma [45], and *LRP1B* has been proposed as a single gene surrogate to this status (mutational load, in other words, [46]), which correlates with response to immunotherapy. Patients with heavily pretreated castrate-resistant prostate carcinoma carrying *LRP1B* mutations had significantly better (75%) and longer response to anti-PD-1 antibody pembrolizumab than non-carriers (14%) [47]. Our data on PDAC are in agreement with literature on other cancers, suggesting that *LRP1B* mutations may mark high-risk cancers prone to generalization on one side and suggest targeted therapy for those already generalized on the other side.

For three patients, samples of primary tumors and paired pulmonary metastases were available for comparison. Both overall mutational load and mutation spectra of individual genes markedly differed between pairs, suggesting that adjuvant chemotherapy after resection of the untreated primary tumor could create “selective bottlenecks” and, thus, change mutation spectra in metastatic lesions [48]. These changes have yet to be considered for treatment decisions. Interestingly, the mutation profile of major PDAC driver *KRAS*, amended with ddPCR data, agreed between both tissue types.

Some other, particularly interesting, results were observed through comparison of primary tumors with metastases in the present study. *FLG2* (filaggrin 2, OMIM no. 616284) was mutated in all studied primary tissues, but in none of the paired metastatic lesions. Together with the very low level of its expression in lung carcinomas and moderate expression in pancreatic adenocarcinoma, reported previously (www.proteinatlas.org), this observation requires further research. Present study results also correspond with the previously reported role of *FLG2* mutations in early stages of carcinogenesis as they have been found in precancerous lesions but not in gastric adenocarcinoma or peritoneal gastric cancer metastases [49]. *FLG2* was recently listed among “untouchable genes” with a deficit of loss-of-function mutations, suggesting that these genes may provide tumor cells with a survival advantage and eventually serve as alternative therapeutic targets [50]. The present study strengthens this assumption and provides a lead for subsequent studies aimed to control the PDAC metastasis process.

Complementing sequencing data with ddPCR assessment of the most common *KRAS* G12V variant improved the detection rate of overall *KRAS* mutation frequency from 60% to 80%, a rate comparable to the previously published data [10,11,12,13,51]. This also demonstrates the necessity to consider the use of techniques with higher sensitivity over conventional next-generation sequencing, including the publicly available whole-exome or -genome profiles, for assessment of key drivers in cancers. Although visual inspection of raw data with IGV was able to improve the detection rate by the next-generation sequencing method, such inspection is not routinely performed and cannot be considered accurate without further confirmation, e.g., by ddPCR.

Our study has some limitations and benefits. First, the modest sample size may be seen as a limitation. This was unavoidable as a single center does not have access to large numbers of surgically treated PDAC patients with both primary and distant pulmonary metastatic lesions available for research. Alternatively, a multicentric design could bring more heterogeneity in sample processing and storage, surgical procedures, and follow-up of patients. On the other hand, long-term follow-up of patients enabled the present study to address differences between short- and long-term survivors of PDAC. Considering the published differences in median survival between patients with solitary pulmonary metastases (81.4 months), multi-organ metastases (23.4), and non-pulmonary metastases (15.8) [9], it is intriguing to study the genetic background of long survival enabled by the present study. Moreover, collections of patients with metachronous pulmonary metastases, which developed after radical resection of primary tumors, are extremely rare. Second, FFPE samples are known to be a source of sequence bias, especially overestimation of C > T transitions (n.b., these are manifested as C > T and G > A in the Human Genome Variation Society (HGVS) syntax depending on the strand orientation of the particular gene) due to the formation of cross-linking by-products during long-term storage [52]. Although several studies attempted to deal with this fact [53,54], it remains a currently unresolved issue [55] and the process of DNA isolation and further analysis must be carefully optimized for each application [56,57]. In the present study, sequencing artifacts were removed by the FilterMutectCalls tool (part of the GATK pipeline) that calculates orientation bias using a read orientation model. At last, the gene panel instead of the whole exome or genome was assessed. Given the modest sample size and expected mutation frequencies, highly mutated genes in PDAC and LUAD (target tissues of both inspected lesions) complemented with the frequently mutated pan-cancer driver, and druggable or clinically actionable genes were studied to allow comparison of small subgroups of patients divided by survival and metastatic site. Whole-genome or -exome sequencing approach in larger sample sets will be needed to evaluate the importance of rare somatic variants, including copy number variations.

## 5. Conclusions

In conclusion, this study provides a unique somatic mutation profile of a rarely studied subgroup of PDAC patients and contributes to the concept of precision medicine of this cancer.

## Figures and Tables

**Figure 1 genes-11-01391-f001:**
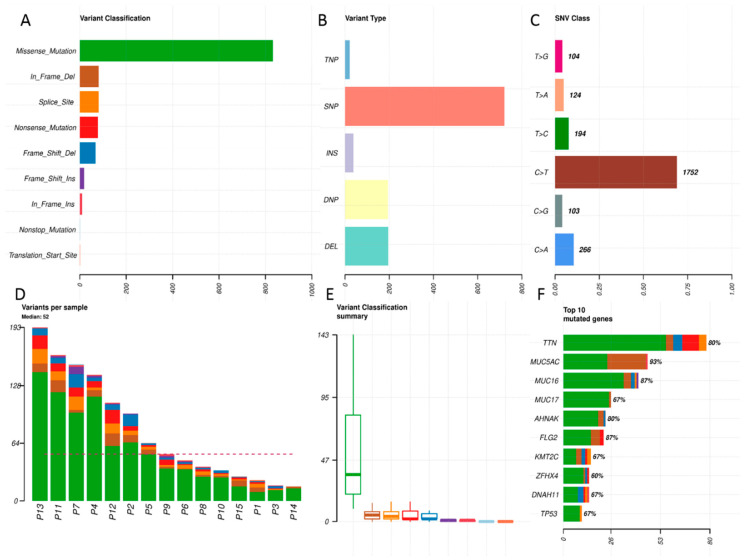
The summary of the distribution of variants in primary tumors. This figure shows the overall distribution of the variants. Only protein-changing variants were considered and 3412 synonymous variants were excluded from the analysis. (**A**) The classification of variants according to the functional effect with counts on the *x*-axis. Most of the variants found were missense mutations. (**B**) The types of the variants. (TNP stands for trinucleotide variant; SNP, single nucleotide polymorphism; INS, insertion; DNP, dinucleotide variant; DEL, deletion.) (**C**) The type of nucleotide change. Note that the most abundant is C > T transition typical for the formalin-fixed paraffin-embedded (FFPE) origin of the samples. A caution in the interpreting of the data due to the paraffinization artefacts is needed. (**D**) The counts and distribution of the variants for the indicated samples; dashed line represents a median (52 variants per sample). Color coding corresponds to the classification of variants in top left. (**E**) The box and whisker plots of the variant classes recognized by the color coding. (**F**) Top 10 genes with the highest counts of the variants on the *x*-axis). Percentages of patients harboring any variant in the indicated genes are shown at the top of the bars.

**Figure 2 genes-11-01391-f002:**
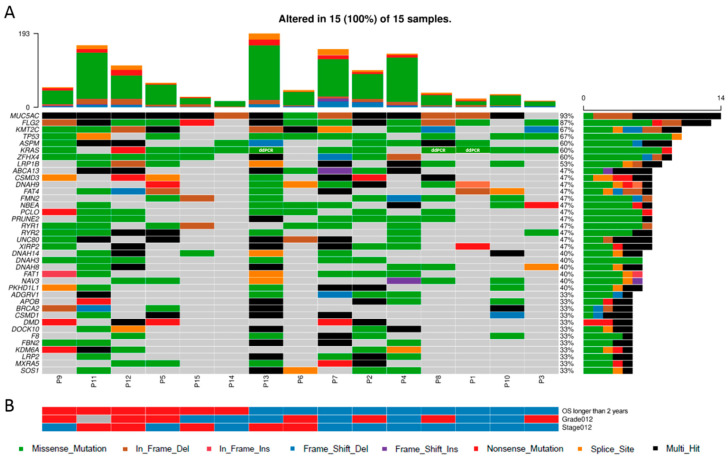
Oncoplot of genes with highest counts of variants in primary tumors. (**A**) Top 50 genes that contain any variant in the highest percentage of the patients displayed. Note that 12 genes were excluded from the analysis because of the presence in the FrequentLy mutAted GeneS (FLAGS) list (Appendix A). The digital droplet polymerase chain reaction (ddPCR) results are included and highlighted in white letters. (**B**) Clinical features of the patients. Patients were sorted according to the overall survival. Patients surviving longer than two years after diagnosis are depicted in blue color and are placed in the right part of the plot. Patients with shorter survival than two years are depicted in red. Well to moderately differentiated tumors (histologic grade G1 or G2) and tumors with less advanced stage (pathological stage I or IIA) are depicted in blue. Poorly differentiated tumors (grade G3) and tumors with more advanced stage IIB are depicted in red. Histological grade was not available for one sample (depicted in gray).

**Figure 3 genes-11-01391-f003:**
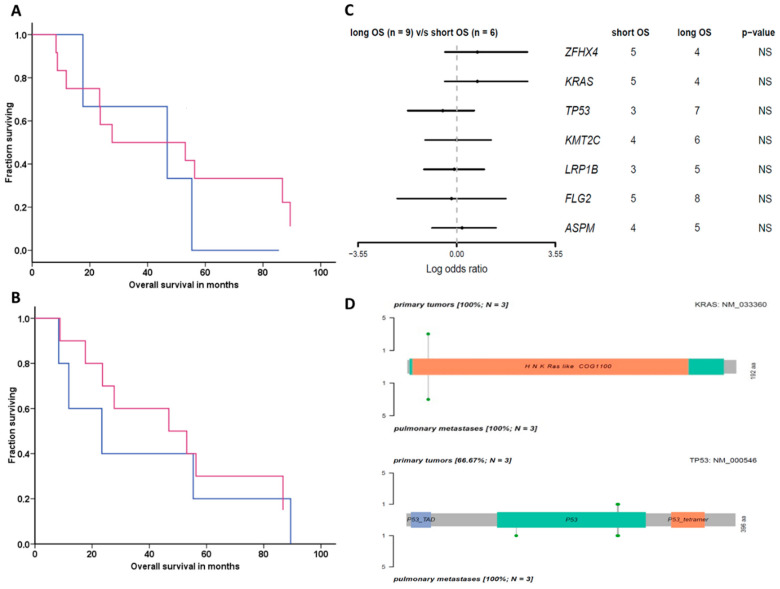
Association of *KRAS* and *TP53* mutations with survival and metastases. (**A**) Associations of *KRAS* and (**B**) *TP53* mutations with survival: red lines for mutated and blue lines for wild type. (**C**) Forest plot showing odds ratios of genes conferring shorter overall survival. (**D**) Lollipop plot comparison of mutations in primary tumors and metastases: *KRAS*, the longest transcript (192 amino acids) is shown (left); *TP53*, the longest transcript (396 amino acids) is shown (right). Left axes show the number of samples with mutation hit. Missense mutations are depicted in green points.

**Figure 4 genes-11-01391-f004:**
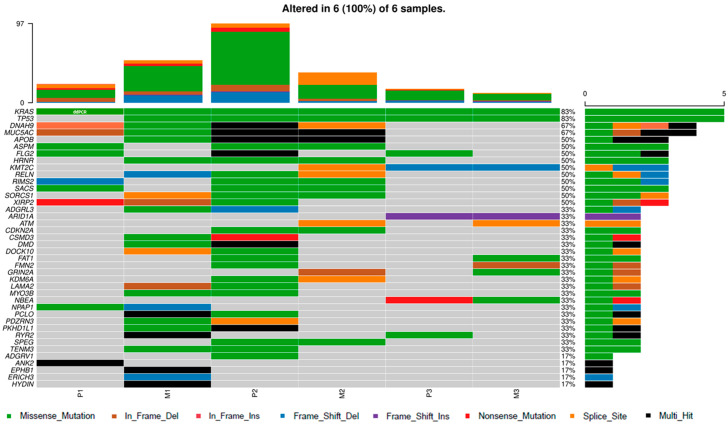
Oncoplot of genes with highest counts of variants in primary tumors and matched pulmonary metastases. Top 50 genes that contain any variant in the highest percentage of the patients are presented. Note that 10 genes were excluded from the analysis because of the presence in the FLAG list (Appendix A). The ddPCR results are included and highlighted in white letters. Samples are ordered in pairs of primary tumors (P) and metastases (M).

**Figure 5 genes-11-01391-f005:**
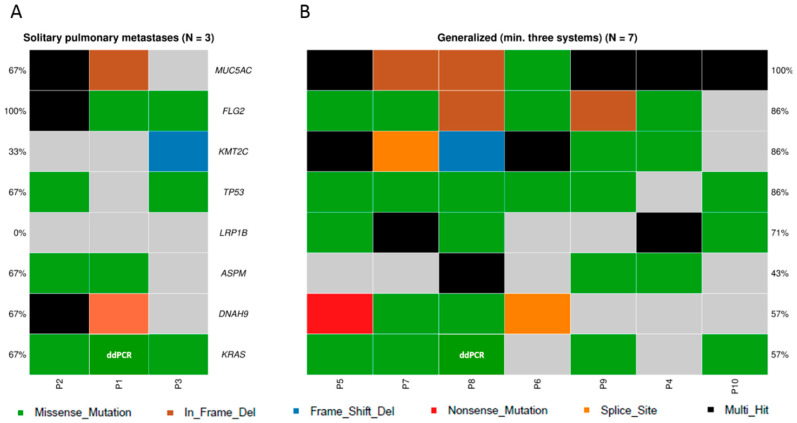
Oncoplot of genes with highest counts of variants in primary tumors. Comparison between groups of (**A**) patients with solitary pulmonary metastases and (**B**) patients with generalized disease. Top five mutated genes in both groups are shown. Genes in the FLAGS list (Appendix A) were not used for the analysis. The ddPCR results are included and highlighted in white letters.

**Table 1 genes-11-01391-t001:** Clinical characteristics of the patients.

Parameters	Number of Patients	%
Age at diagnosis, mean ± SD (years)	62.1 ± 10.7	100
Sex		
Male	5	33
Female	10	67
TNM stage		
I	3	20
IIA	7	47
IIB	5	33
III	0	0
IV	0	0
Histologic grade (G)		
G1	1	7
G2	6	43
G3	7	50
Gx	1	0
Angiolymphatic invasion (pL)		
pL0	7	47
pL1	8	53
Perineural invasion (pP)		
pP0	2	13
pP1	13	87
Angioinvasion (pA)		
pA0	11	73
pA1	4	27
Adjuvant therapy		
Gemcitabine	11	79
None	3	21
Unknown	1	0

**Table 2 genes-11-01391-t002:** *KRAS* G12V allele frequencies in primary tumors and metastases of PDAC patients.

Sample Number	Allele Frequency (%) ddPCR	Allele Frequency (%) Sequencing	Sequencing Results
P1	0.66 ± 0.11	ND ^2^	wild type ^4^
P2	6.02 ± 0.11	6.80	confirmed
P3	5.86 ± 0.08	6.50	confirmed
P4	BLQ ^1^	ND ^2^	wild type–confirmed
P5	0.25 ± 0.09	ND ^2^; 10.60 (G12D)	wild type for G12V ^4^
P6	BLQ ^1^	ND ^2^; 6.00 (G12D)	wild type for G12V–confirmed
P7	10.70 ± 0.57	37.00	confirmed
P8	3.63 ± 0.04	6.52 ^3^	wild type ^3^
P9	15.60 ± 0.06	13.10	confirmed
P10	BLQ ^1^	ND ^2^; 8.60 (G12D)	wild type for G12V–confirmed
P11	BLQ ^1^	ND ^2^	wild type-confirmed
P12	2.82 ± 0.38	1.79 ^3^	wild type for G12V ^3^
P13	2.26 ± 0.11	1.83 ^3^	wild type ^3^
P14	7.13 ± 0.23	9.00	confirmed
P15	BLQ ^1^	ND ^2^; 7.90 (G12D)	wild type for G12V–confirmed
M1	19.04 ± 0.46	17.30	confirmed
M2	20.27 ± 0.65	20.90	confirmed
M3	5.41 ± 0.46	6.10	confirmed

^1^ BLQ = below limit of quantification, wild type. ^2^ ND = not detected, wild type. ^3^ Manually confirmed in the Integrative Genomics Viewer Appendix A). ^4^ Discordant results compared to ddPCR (*n* = 2).

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
