# Peer review of "Targeted Sequencing of Pancreatic Adenocarcinomas from Patients with Metachronous Pulmonary Metastases"

_genes, 2020, doi:10.3390/genes11121391_

Round 1

Reviewer 1 Report

The presented study contains an in depth study of a small sample of patients. While high powered multi center studies are important, so are small studies of single centers. Further, there is a demographic advantage as multicenter studies sometimes exceed single populations that may have endemic problems and aetiologies. Nevertheless, particularly the PDAC centric section of this paper could benefit from an additional inclusion of the TCGA dataset possibly with a compare and contrast section between this small dataset and which aspects hold true and may not hold true compared to inclusion of the TCGA patients with a similar background. I assume the pipeline as performed stands. This could significantly strengthen the authors conclusions on the difference between OS given that with this dataset the authors found no significant differences. Inclusion of another 10-20 patients in each group could be critical!

Further, the title concentrates on secondary pulmonary metastases which as far as I can see represents a subset of the analyzed data, the main and slightly higher powered part (9 vs 6 patients as opposed to 3 patients with pulmonary metastases) being OS > 2 vs OS < 2 years. These two analyses could be clearer worked out especially in the introduction and representation in the title.

1) It would be helpful to be more exact in the introduction. line 44 - what type of subgroup; line 53-57 surgically resected PDAC suggests resection of the pancreas whilst the previous section suggests that your interest lies in pulmonary metastases. Without reading on I am now confused what the paper is about. After reading the paper I have to say that the introduction could be expanded with respect to background for the critical sets of data and especially with respect to the last paragraph lines 53-57 explaining what is contained in the rest of the paper.

2) Please go through he paper and check abbreviations which should fist be introduced (OS, pTNM staging criteria, FFPS, LUAD, et cetera).

3) From skimming section 2.6 it seems to me that the blood of patients was used as a control to establish mutations in the tumor tissue. What about heritable mutations? Would these not be filtered out of the analysis this way?

4) The authors discuss long-term survival of 5 years as a strength of their paper yet in the text use a criterion of 2 years as lng-term survival. Why this discrepancy?

5) The title is misleading. The title suggests a decent dataset of patients with secondary pulmonary metastases. Really the number here is 3 and the pulmonary metastases is a small subgroup of the already small study group.

6) The second point to the pulmonary metastasis the authors developed in their results concerns the length of survival with respect to somatic mutations. This is a very interesting point. However, the fact that we are dealing with 15 patients remains. I am unsure as to whether TCGA cancer data has been looked at in this perspective. However, glancing at the PDAC TCGA datatable it appears that there are 8 deceased patients that lived >720 days and 17 living patients that were followed up >720 days. Together with deceased patients < x days possibly excluding unknown causes of death or surgical complications the authors could surely use their pipeline to power this aspect of the paper significantly better.

Reviewer 2 Report

In this manuscript the authors perform targeted sequencing in DNA from blood and archival samples of 15 primary tumors and three paired metastases 15 pancreatic cancer patients. This was coupled with blood analysis of G12V mutation in KRAS by DD-PCR. Majority of aberrations were missense mutations, SNPs and common nucleotide change was C>T. The individual gene frequencies and mutation frequencies did not correlate with survival. Average somatic mutations were 254 of which 52 had an impact on protein. Some genes did stand out such as LRP1B found mutated in generalized malignancy. FLG2 mutation was exclusive to primary tumors. Although mentioned in the shortcomings the sample size remains small. As such the study does show a unique somatic mutation profile in pdac patient that have metachronous pulmonary metastases.

Figures are not labelled properly. Panels within each figure should be labelled (a,b,c,d) and mentioned in text to make it simpler to follow

Discussion section mentions 5 yr survival at 5%. Although latest figures put it around 10% so it should be corrected, and discussion should be aligned around this new figure

Some typos were found in the text that should be checked and corrected
